# Modelling smallholder farmers' preferences for soil fertility management technologies in Benin: A stated preference approach

Segla Roch Cedrique Zossou[1]☯*, Patrice Ygue Adegbola[2]☯, Brice Tiburce Oussou[2]☯, Gustave Dagbenonbakin[2]☯, Roch Mongbo[3]☯

1 International Center of Research and Training in Social Science, Abomey-Calavi, Benin, 2 National Agricultural Research Institute of Benin, Cotonou, Benin, 3 Faculty of Agricultural Sciences of University of Abomey-Calavi, Cotonou, Benin

☯ These authors contributed equally to this work.
* rochybuggs@yahoo.fr

**Data Availability Statement:** All relevant data are within the manuscript and its Supporting Information files.

## Abstract

The decline of soil fertility is a major constraint which results in lower levels of crop productivity, agricultural development and food security in Sub-Saharan Africa. This study is the first to perform a focalized investigation on the most interesting technological profiles to offer to each category of producers in Benin agricultural development hubs (ADHs) using the stated preference method, more precisely, the improved choice experiment method. The investigation focused on 1047 sampled plots from 962 randomly selected producers in villages of the Smallholder Agricultural Productivity Enhancement Program in Sub-Saharan Africa of the ADHs. An analysis of the experimental choice data with the endogenous attribute attendance and the latent class models was carried out to account for the attribute non-attendance phenomenon and the heterogeneity of the producers' preferences. However, three classes of producer with different socio-economic, demographic, and soil physicochemical characteristics were identified. Thus, the heterogeneity of preferences was correlated with the attributes linked to the cost, sustainability, and frequency of plot maintenance. All producers, regardless of the ADHs, had a strong attachment to accessibility of technologies with short time restoration of soil fertility, and the ability to obtain additional benefits. These latest attributes, added to that relating to cost, tended to have a low probability of rejection in the decision-making process. These results have implications for local decision-makers facing the complex problem of resolving land degradation and local economic development challenges. The generalizability of these findings provides useful insight and direction for future studies in Sub-Saharan Africa.

## Introduction

Agriculture is the key to ensuring food security [1]. However, despite its importance, agriculture has not yet reached its potential in emerging economies because it is subject to several land, social, and economic constraints. It is therefore exposed to high rainfall dependence, low

**Funding:** We gratefully acknowledge and thank The Smallholder Agricultural Productivity Enhancement Program in Sub-Saharan Africa (https://ifdc.org/) for their financial support for this research. The Smallholder Agricultural Productivity Enhancement Program in Sub-Saharan Africa was not involved in the research or development of this project.

**Competing interests:** The authors have declared that no competing interests exist.

level of use of technological innovations, and low level of education among its stakeholders [1]. In addition, one of the reasons often attributed to the lack of agricultural research performance in Benin, and Sub-Saharan Africa in general, is related to land and ecosystem degradation [2–4].

Africa loses 8 million metric tons of soil nutrients each year and 95 million ha of land has been degraded to the point of significantly reducing productivity [3,5]. This situation is explained by high population growth and the intensification of land use with the direct consequence of reducing the fallow period [6], decreasing land productivity, and increasing charges to maintain crop production levels. Steiner [7] showed that the direct consequence is the threat to food supply. A research carried out in Benin shows that the loss of major nutrients from the soil is very high and far exceeds the inputs (mineral fertilizers in particular), resulting in a negative balance of N, P, and K minerals in particular [6,8]. These losses are mainly due to exportation through crops, water erosion and leaching, and mining practices [9].

In response to this situation and the continuing need to increase agricultural production, a set of strategies and technologies for soil fertility improvement in general and agroforestry in particular, which could help producers to improve crop yields and limit pressure on forest resources, have been developed through agricultural research and disseminated over the years. In Benin, several strategies have been developed by farmers over several centuries to maintain and restore soil fertility [10]. However, the existing body of literature support that other strategies (agronomic, biological, zootechnical, agricultural land use planning, and combined practices) have been introduced by research and development structures [4,11,12]. Researchers have strongly argued that despite the actions of the government and technical and financial partners to support the dissemination and adoption of these technologies, the problem of declining soil fertility persists [4,12]. Besides mineral fertilizers, these new technologies have been poorly implemented. Several previous studies have highlighted the low level of implementation of these technologies by producers despite their technical performance [2,4]. This situation may be linked to the incompatibility of the characteristics of these new technologies with those desired by farmers [4]. Moreover, these previous studies do not reveal the heterogeneity of producers' preferences for soil fertility management technologies.

The producers' willingness to accept new technologies will increase through access to better technologies that fit their individual characteristics [13–16]. These characteristics are related to the socioeconomic, demographic, farm, and soil characteristics of the producers. Thus, this study contributes to the literature by including the physicochemical and biological characteristics of producers' soil in the sources of heterogeneity of preferences by considering producers' plots as an experimental observation unit.

This study offered the persistent effort to address the gaps and limitations identified from the previous studies. To better involve stakeholders in the field and enable them to adapt their practices, this study assessed the technological profiles appropriate for producers' plots, using the stated preference (SP) assessment method. The SP method is usually based on random utility theory and relies on assumptions of economic rationality and utility maximization [17]. To operationalize this approach, several data collection methods have been developed. These are the pairwise comparison approach, traditional full profile approach, adaptive or hybrid joint approach, and experimental choice (EC) or discrete choice approach [18,19]. The first three approaches are difficult and require respondents to have high intellectual skills unlike the EC approach, which is much easier [19,20]. In addition, the experimental choice method is very popular and provides a solid basis for stated preference studies by providing a measure of the benefits sought by different categories of individuals [19,21,22]. This method has been used in consumer choice studies to understand how consumers make their purchasing decisions, predict their behaviour, and determine the value and importance of different characteristics of

goods to offer them products that correspond well to their requirements [23,24]. The experimental choice method proposes to the respondent to make a choice from a set of several alternatives or profiles. It is closer to the real purchase or adoption process of the economic agent, and therefore, better represents real market situations [21]. The latter approach is the one used in the present study.

Our study is part of a process to further develop all of the above-mentioned work using improved econometric models, the latent class logit model (LCM), allowing for the heterogeneity of preferences to be taken into account [25,26]. This choice of modelling highlights the expectations of different classes of producers and their willingness to adopt for attributes. The attribute non-attendance (ANA) was particularly taken into account by using the endogenous attribute attendance econometric model (EAA), which constitutes progress compared with previous experimentation studies on choice [27,28]. In addition, the present study was carried out following the new conformation of the agricultural sector. To this end, it provides updated data in line with current agricultural policy, which involves agricultural development hubs (ADHs).

This study should guide the actions of public and private decision-makers, as well as those of NGOs, in the development or dissemination of appropriate technologies per ADH. This manuscript is structured according to the following plan. The following section presents the materials and experimental selection methods as well as the empirical model used. The third section presents the estimated results, which are then discussed in Section 4. The final section presents the conclusions and implications for the development and a widespread adoption of soil fertility management technologies in Benin.

## Materials and methods

### Study zones

The study was conducted in 2017 in Benin, West Africa, more specifically, in 18 villages of the intervention zones of the Smallholder Agricultural Productivity Enhancement Program (SAPEP) in Sub-Saharan Africa. Villages of the intervention zones of the SAPEP, were randomly selected during a baseline survey, taking into account the agricultural production areas and the presence and levels of production in the promising sectors targeted by the project. Subsequently, these villages were divided into ADHs to spatially characterise the demand in terms of soil fertility management technology. The villages explored for the present study are illustrated in Fig 1.

### Selection of producers and sampling of plots

The study was conducted not only with the heads of producers' households that were previously randomly selected during the baseline study conducted by the project but also with an individual of the opposite sex to the head of the household to compare intra-household perceptions. Thus, 962 producers were surveyed (51.61% male, 48.39% female).

The producers' plots were the observation units for soil sampling and experimental technology selection. Soil samples were taken from the producers' plots to assess the physicochemical and biological characteristics of the soil. The study was carried out on private land. We confirm that, the owner of the land gave permission to conduct the study on this site, and no specific permissions were required for these locations/activities. The agents of the research centers, the town hall, the village chiefs, and the resource persons facilitated the interviews with the producers in the villages. The identification of these characteristics facilitates the determination of the preferences of producers' segments with particular soil characteristics. Given the large size of each producer's plots, the total number of plots sampled ($N$) was

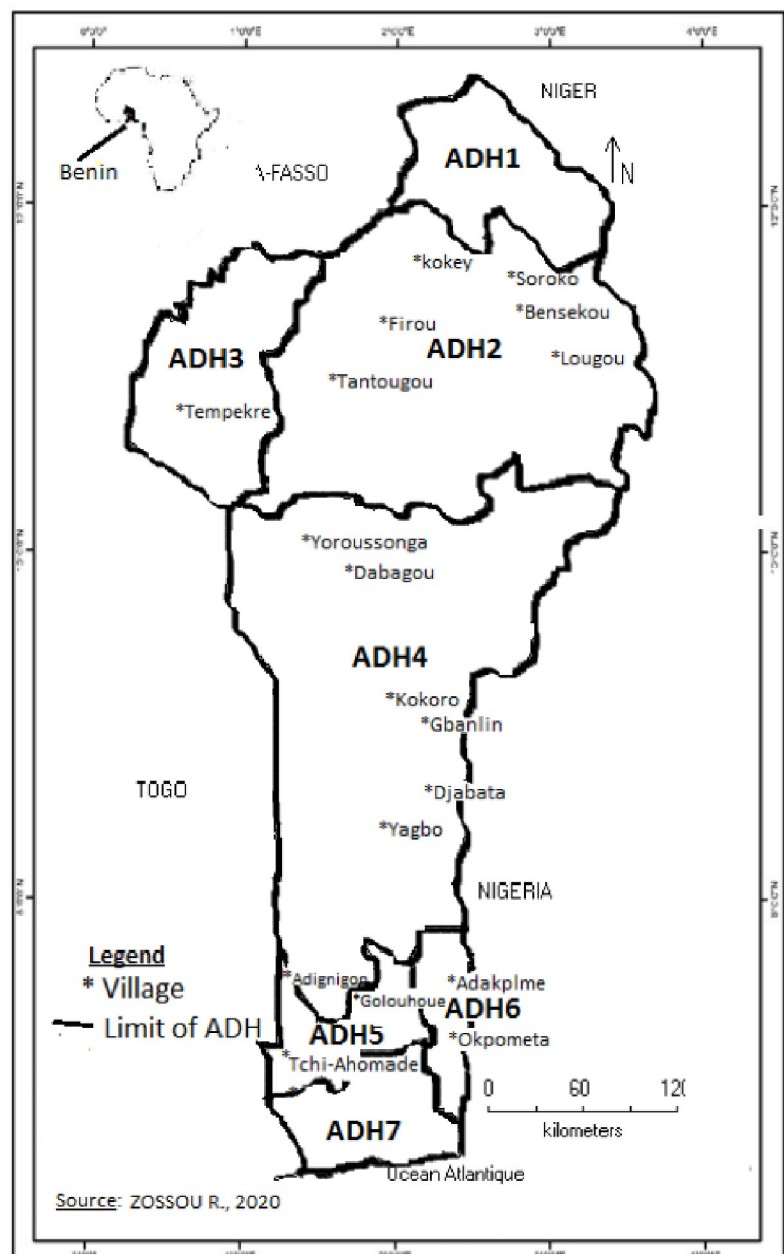

**Fig 1. Study zone.**

determined according to the following formula of Glele-Kakai and Sinsin [29]:

$$N = \frac{T_{1-\alpha/2}^2 Cv^2}{d^2}$$

(1)

where $N$ is the total number of plots to be explored; $t_{1-\alpha/2}^2$ ($\alpha = 5\%$) is the critical value of the Student distribution t that converges to the normal distribution for large samples ($N > 30$) and is equal to 1.96; $Cv$ is the coefficient of variation of the number of producers' plots in the villages considered. It is equal to 51.26% (baseline survey); and $d$ is the margin of error set at

5%. Thus, a total of 1047 parcels were investigated. It should be noted that the sample plots were randomly selected from those of producers while ensuring diversity in integrated soil fertility management practices. Note that soil samples were collected on the sampled plots producers to assess the physico-chemical and biological characteristics of soils. These composite soil samples weighing around 500 grams were taken 20 cm deep.

### Assessment of soil parameters to determine their level of limitation

The composite soil samples taken were sent to the INRAB Water and Environment Soil Science Laboratory for analysis.

Sys et al. [30] agreed five degrees of intensity of limitations for soil parameters:

- Degree I: no limitation, the characteristic of the soil considered is optimal;

- Degree II: slight limitations, referring to situations that could slightly decrease yields without however imposing special cultivation techniques;

- Level III: moderate limitations, referring to situations which cause a greater reduction in yields or the implementation of special cultivation techniques. These limitations do not affect profitability;

- Level IV: severe limitations, referring to situations which cause a reduction in yields or the implementation of cultivation techniques which could jeopardize profitability;

- Degree V: very severe limitations, referring to situations that no longer allow the use of land for the specific purpose.

Thus, the degrees of intensity of the limitations of the chemical characteristics, for each of the sampled plots of the producers were determined by following the parameters defined by Sys et al. [30]. Table 1 summarizes the criteria for evaluating the degrees of limitation of soil chemical parameters, defined by Sys et al. [30].

### Experimental choice method

The EC method is usually based on random utility theory and relies on assumptions of economic rationality and utility maximization [17,31]. In random utility models, the probability of observing a specific realization of a choice to be modelled is determined according to a decision rule formulated in terms of latent or unobservable variables that are associated with it

Table 1. *Criteria for assessing the degrees of limitation of soil chemical parameters.*

| Soil chemical parameters | Degrees of limitation | | | | |
|---|---|---|---|---|---|
| | Degree I (Without limitations) | Degree II (Weak limitations) | Degree II (Weak limitations) | Degree: IV (Severe Limitations) | Degree: V (Very severe limitations) |
| Organic matter | > 2 | 2–1.5 | 1.5–1 | 1–0.5 | < 0.5 |
| Total nitrogen | > 0.08 | 0.08–0.06 | 0.06–0.045 | 0.045–0.03 | < 0.03 |
| P ppm (Bray $_1$) | > 20 | 20–15 | 15–10 | 10–5 | < 5 |
| K (meq/100 g of soil) | > 0.4 | 0.4–0.3 | 0.3–0.2 | 0.2–0.1 | < 0.1 |
| Sum of exchangeable bases (meq/100 g of soil) | > 10 | 10–7.5 | 7.5–5 | 5–2 | < 2 |
| Base Saturation (V) | > 60 | 60–50 | 50–30 | 30–15 | < 15 |
| CEC (meq/100 g of soil) | > 25 | 25–15 | 15–10 | 10–5 | < 5 |
| pH | 6.5–6.0 | 6.0–5.5 | 5.5–5.3 | 5.3–5.2 | < 5.2 |
| | 6.5–7.8 | 6.5–7.8 | 7.8–8.3 | 8.3–8.5 | > 8.5 |

[32]. Thus, the indirect utility (*U*) that the producer (*i*) receives from the given attributes for an alternative (*j*) takes the following linear form:

$$U_{ij} = V_{ij} + \varepsilon_{ij}, \qquad j = 1, \ldots \ldots K \text{ and } i = 1 \ldots \ldots \ldots n \qquad (2)$$

where *Vij* is the deterministic (non-stochastic) part of the utility and $\varepsilon_{ij}$ is the random (or stochastic part consisting of the error terms of the model) that takes into account uncertainty. Depending on the assumptions adopted to represent the distribution of the random portion, different discrete choice models can be distinguished. However, mathematically, discrete choice models are generally based on the assumption that the choice probabilities relating to the utility function can be estimated by the multinomial logit model (MLM). However, this model has limitations regarding to the assumption (Gumbel's law) of identically distributed independence (IDI) of error terms between alternatives and observations, and therefore assumes homogeneity of preferences [33,34]. Another limitation of the MLM is related to the assumption of independence of irrelevant alternatives (IIA). It is the capital limit of multinomial logit [35]. To overcome these limitations, several other alternative models are available. These are the nested logit, crossed nested logit, latent class model (LCM), polytomic probit model, mixed logit, and generalized multinomial logit [34].

The nested logit first proposed by [35], which is also part of the same family of generalized extreme values as the multinomial logit, does not allow us to completely avoid the IDI and IIA hypotheses. More flexible than previous models, the polytomic probit is not constrained by the three previously developed limits. However, the estimation of this model generates too heavy econometric calculations. The mixed logit model (MXL) is not constrained by any of the above limitations and can detect possible heterogeneity not observed in preferences [34]. The latent class logit model (LCL) also does not violate the IIA hypothesis and differs from the MXL model in that it allows the distribution of coefficients to be discrete rather than continuous. It uses a statistical methodology based on the concept of likelihood to identify sources of heterogeneity at the segment level rather than at the individual level as does mixed logit [26]. It can be considered to be an improvement of the MXL. This model has been increasingly applied in recent segmentation studies and has produced promising results [25,26]. Several authors have compared the two approaches (MXL and LCL). Some concluded that each approach has its own merits and even that the LCL is more efficient in terms of estimation [36]. Others, such as Scarpa et al. [37], explain that the LCL has the advantage of being based on a joint estimate and that it allows a more intuitive interpretation, facilitating communication with decision-makers.

In addition, more recent studies have revealed that respondents participating in discrete choice experiences often ignore certain attributes in their decision-making process (called ANA). Resulting biases in modelling occur when these aspects are not taken into account [28,38,39]. Two approaches were proposed to account for the ANA. These include stated ANA and inferred ANA. Stated ANA is an experimental approach, which consists of asking the respondent to answer specific questions about the attribute ignored during decision-making [24]. Inferred ANA is an econometric approach that provides a better fit for the model, while the stated ANA is not consistent[27,28]. First, the fact that respondents attribute low importance to certain attributes, which could be ignored at the beginning of their choices, leads to overestimation [28]. In addition, the answers to questions relating to the ignored attributes can be a source of potential problems regarding endogeneity bias [27]. LCL and EAA are econometric models widely used to account for inferred ANA models [38,40]. For these reasons, LCL and EAA were applied in the present study for data analysis.

**Latent class logit model.** The basic assumption of the LCL model is that an individual belongs to a specific segment but that the members of each segment are unobservable. As a result, respondents from different segments will have different preferences. The LCL simultaneously estimates the probability that a producer will select a given technology from the set of choices and belongs to a specific segment [25]. Let us consider producer $i$ who selects alternative $j$ from $K$ technology alternatives in the set of choices. Furthermore, assuming that he/she belongs to segment s, where $s \in S$, the indirect utility function for his/her preferred technology profile $j$ is written as follows:

$$U_{ij/s} = X_{ij}\beta_s + \varepsilon_{ij/s} \tag{3}$$

where $X_{ij}$ is a vector representing the attributes concerning to the $K$ alternatives and $\beta_s$ is the parameter vector of the segment $s$ associated with the vector $X_{ij}$ and $\varepsilon_{ij/s}$ as error terms. Assuming that the error terms are independently and identically distributed (IID) and follow the Gumbel distribution, the probability that $i$ will select an alternative among the $K$ alternatives while belonging to a given segment is:

$$P_{K/s} = \prod_1^k \frac{\exp(Xij\beta s)}{\sum_1^k \exp(Xik\beta s)} \tag{4}$$

Let us now consider $M_{ns}^* = \lambda_s Z_n + \xi_{ns}$ a function of the probability ($P_s$) belonging to a segment $s$ among the $S$ unobservable segments where Z represents individual characteristics of the producer and λ_s (s = 1, 2, 3...S) represents the parameters to be estimated specific to each segment. Assuming that the term of error $\xi_{ns}$ is independently and identically distributed between producers and segments and follows a Gumbel distribution, then the probability that he/she belongs to segment $s$ can be expressed as follows:

$$P_s = \frac{\exp(\lambda_s Z_n)}{\sum_1^S \exp(\lambda_S Z_n)} \tag{5}$$

By combining Eqs (4) and (5), we obtain the following expression which represents the probability that producer $i$ belongs to segment $s$ and selects technology profile $j$:

$$P_{jn/s} = \sum_1^S P_s P_{K/s} = \sum_1^S \left( \frac{\exp(\lambda_s Z_n)}{\sum_1^S \exp(\lambda_S Z_n)} \prod_1^k \frac{\exp(Xij\beta s)}{\sum_1^k \exp(Xik\beta s)} \right) \tag{6}$$

The log likelihood function to obtain the parameters $\lambda_s$ and $\beta s$ is given by the expression:

$$L = \sum_k \sum_n I_i ln P_{jn/s} \tag{7}$$

where $I_i$ is an indicator variable of the observed choice.

**Endogenous attribute attendance model (EAA).** In the EAA model, each choice is considered a two-step process in which the decision maker first decides which attributes to take into account when comparing profiles. Then, he selects the profile with the best characteristics, taking into account his individual characteristics [28,38]. Thus, the formulation of the basic logit of EAA is presented as follows:

$$P_{ijs|C_K} = \frac{e^{\sum_{k \in C_k} \beta_i^k X_{ijt}^k}}{\sum_{j=1}^J e^{\sum_{k \in C_k} \beta_i^k X_{ijt}^k}} \tag{8}$$

where $X_{ijt}^k$ represents the individual ($i$) who selects the modality of the attribute ($k$) relative to

the profile ($j$) of the subset of choice attributes $C_k$, in the choice situation (t); and $\beta_i^k$ denotes the specific coefficient for the attribute ($k$).

According to [41], the probability that individual i takes into account attribute k is specified by $\frac{e^{\gamma_k Z_{ik}}}{1+e^{\gamma_k Z_{ik}}}$, where z is a vector of individual characteristics and $\gamma$ is a vector of parameters to be estimated [28,42]. Assuming that these probabilities are independent of attributes, the probability of choosing a profile designated as a subset of attributes ($C_k$) is given by:

$$P_{iC_k} = \prod_{k \in C_k} \frac{e^{\gamma_k Z_{ik}}}{1 + e^{\gamma_k Z_{ik}}} \prod_{k \in C_k} \frac{1}{1 + e^{\gamma_k Z_{ik}}} \tag{9}$$

The probability that individual i selects profile j from the set of choices (C) in a given situation (t) can be written as follows:

$$P_{ijt}^{EAA} = \sum_{k=1}^{K} P_{iC_k} \prod_{t=1}^{T} \prod_{j=1}^{J} \left( P_{ijt|C_k} \right)^{Y_{ijt}} \tag{10}$$

where $Y_{ijt}$ takes the value 1 when option (j) is chosen, and 0 otherwise; $f(\beta_n|\theta)$ denotes the density for $\beta_n$ in which $\theta$ is the distribution parameter.

## Experimental design

Since the EC technique is characterized by a statistical design of hypothetical alternatives [43,44], soil fertility management technology profiles were composed based on the main choice attributes reported by users during the exploratory phase. This first phase of the survey took place in 2017 through focus groups and using an interview guide developed on the basis of previous studies. Thus, analysis of the data from this phase using the Kendall method made it possible to prioritize an exhaustive list of the main attributes of the selection of practices. Consequently, six main attributes were selected, each with two levels, except for the cost level; cost was the monetary attribute.

Restoration time is the duration of soil remediation, which can be fast or slow depending on the technology used. It has been demonstrated previously that the misuse of chemical fertilizers is due to its rapid effect on crop growth. Shrub legumes have a much longer cycle and therefore assume slow soil restoration, unlike herbaceous legumes (e.g. Mucuna) or synthetic chemical fertilizers [3,45].

Accessibility corresponds to the availability of the basic materials involved in the realization of technology. The unavailability of chemical fertilizers has always been raised as a constraint to implementation [46]. Consequently, the massive use of technologies would be owing to their high availability and ease of access.

Regarding the possibility of obtaining additional benefits, some technologies (regeneration based on, e.g., soya, cowpea, and cassava) are appreciated for their ability to restore soil fertility (biomass supply) and increase yield but also facilitate the production of edible or marketable products contributing to increasing, e.g. income, household food security, and livestock feeding [47]. Others (regeneration with, e.g. Mucuna) are less adopted because they do not facilitate the production of additional benefits. Soil fertility retention time corresponds to the effectiveness of the technology over one or more production campaigns following application. The sustainability of innovation is a continuous process of perceiving, which enables business organizations to have new markets, improved products and services [48].

The control expresses the frequency of maintenance of the plot. The attributes that were used during the choice experiment are presented in Table 2 with the associated attribute levels.

Considering the number of attributes and the associated attribute levels, $2^5 \times 5^1 = 160$ possible combinations or theoretical profiles were constructed. It would be very difficult for

**Table 2. Attributes and associated attribute levels.**

| Attributes | Attributes levels |
|---|---|
| Restoration time | 1 = Short<br>2 = Long |
| Accessibility | 1 = Difficult<br>2 = Easy |
| Possibility of obtaining additional benefits | 1 = Impossible<br>2 = Possible |
| Soil fertility retention time | 1 = Temporary (1 production campaign)<br>2 = Long (more than one campaign) |
| Regular control (frequency of maintenance of the plot) | 1 = Less control<br>2 = Regular control |
| Purchase cost CFAF per hectare | 0; 70,000; 100,000; 150,000; 220,000 |

respondents to objectively consider and judge 160 profiles before making a precise choice. To facilitate their choice, the discrete choice sets were restricted to 16 realistic profiles divided into four groups of four profiles each (4 × 4) with the efficiency index Dz estimated at 98.81% [44]. This result was possible in the SAS software package using the experimental design of Street and Burgess [49], which is based on the criterion of optimality of efficient design. Among the profiles thus created, 10 existing practices could be identified by their characteristics and put into play in the choice sets with the other six hypothetical profiles. A reference situation (*status quo*) was added to each set of choices. The different standard technology profiles were presented to each producer in the form of a game. For each set of choices, the respondent was asked to select a technology profile for each of their sampled plots or to select the reference situation that corresponds to their current practice. Each respondent revealed the fictitious situation that provided the most useful information, thus expressing their interest in the attributes according to the profile. To make the exercise easier to understand for the respondents, the different scenarios were illustrated with photos or pictograms accompanied by a brief caption. Examples of the maps proposed to the producers during interviews are shown in Fig 2. The various actors were assured that their virtual choice in the experiment would not have any real immediate consequences on their activities. It was clarified that the results would be used more generally to determine the appropriate technology model for soil fertility management. Data were collected in 2017 using an application installed on tablets that contained a digital version of the survey guide. It should be noted that soil samples were taken from the sampled plots of each producer to assess the physicochemical and biological characteristics of the soil.

## Empirical model

The dependent variable $Y$ corresponds to the choice of a profile preferred by the producer for each of his plots in each set of choices. It takes the value $Y = 1$ if he/she selects a profile and $Y = 0$ for the others that are not chosen. In model specification, the utility that the individual derives from a profile model depends on the main attributes of the technology and some individual characteristics of the producer [13,50,51]. Apart from the main attributes of the technology, the individual characteristics of the producer/household (socio-economic, plot, physicochemical, and biological characteristics), and the production area that can be used for segmentation, are those that could affect the choice or abandonment of fertility management practices [52]. According to the studies carried out by the researchers, potential characteristics are related to gender, formal education, number of active agricultural members, access to credit, number of plots and area planted, fertility level, hubs and sub-hubs of development. Not all of the explanatory variables were included in the LCL estimate. A correlation matrix

| Choice set 2 | | Practice #5 | Practice #6 herbaceous legumes (*Mucuna, Ashynomenae, Stylosanthes*) | Practical #7 Microorganism (*eg. mushrooms*) | Practice #8 Crop rotation | None of these practices interest me, i prefer to maintain current practice (*Option q0*) |
|---|---|---|---|---|---|---|
| **Restoration time** | | Long | Short | Long | Short | |
| **Accessibility** | | **Easy** | Difficult | Difficult | **Easy** | |
| **Possibility of obtaining edible by-products** | | Impossible | Impossible | Possible | Possible | |
| **Soil fertility retention time** | | Temporary (*1 production campaign*) | Temporary (*1 production campaign*) | Long (*more than one production campaign*) | Long (*more than one production campaign*) | |
| **Regular control** (frequency of maintenance of the plot) | | Regular control | Less control | Regular control | Less control | |
| **Purchase cost (CFAF per hectare)** | | **70 000** | **70 000** | **100 000** | **70 000** | |
| *Which of these practices do you choose?* | **Plot1** | | | | | |
| | **Plot2** | | | | | |
| | **Plot3** | | | | | |

**Fig 2. Example of a set of cards proposed during the interview.**

was used to eliminate highly correlated variables and those that did not exhibit variability within the alternatives. Variables such as income were not included in the model because the correlation threshold between the variable and the area planted was very high ($p < 0.01$). As a result, producers with high incomes were those who planted large areas. The current reference situation or practice (*status quo*) was used in the model. The negative sign of the coefficient of this variable would generally imply the motivation of producers to adopt new soil fertility management technologies. Table 3 provides details of the explanatory variables inserted in the model.

Parameters estimated from the latent class logit can also be used to calculate the willingness-to-adopt (WTA) for each attribute, which helps to understand the respondents' motivation and quantify their levels of preference for the attributes. Suppose for attribute $X1$ that the WTA1 wants to be estimated. Estimated parameter $\beta1$ of attribute $X1$ can be interpreted as the marginal utility of this attribute. In addition, let us note by $\delta$ the parameter estimated for the monetary attribute i.e. "the cost", which represents the marginal utility of the currency [41].

**Table 3. Variables used in econometric models.**

| Variable | Modality |
|---|---|
| *Status quo* | 1 = yes and 0 = otherwise |
| **Attributes** | |
| Cost | Continuous variable |
| High restoration speed | 0 = Slow; 1 = Quick |
| Accessibility | 0 = Difficult; 1 = Easy |
| Possibility of obtaining additional benefits | 1 = yes and 0 = otherwise |
| Long conservation life | 0 = Temporary (one production campaign); 1 = Long (more than one campaign) |
| Maintenance frequency (regular) | 0 = Less control; 1 = Regular control |
| Maintenance frequency (regular) × Cost | |
| Accessibility × Cost | |
| **Physicochemical characteristics of the soil** | |
| Organic matter rate (MO) | Continuous variable |
| N | Continuous variable |
| P | Continuous variable |
| K | Continuous variable |
| Soil pH level | Continuous variable |
| Fertility level | Continuous variable (0 = weak; 1 = average; 2 = high) |
| **Socio-economic and demographic characteristics** | |
| Gender | 0 = Woman; 1 = Man; |
| Formal education | 1 = yes and 0 = otherwise |
| Number of active agricultural members | Continuous variable |
| Access to credit | 1 = yes and 0 = otherwise |
| Acreage | Continuous variable |
| Duration of fallow period | Continuous variable |

The WTA associated with attribute $X1$ is given by the following formula:

$$WTA1 = -\frac{\beta_1}{\delta}\ (7)$$

## Results

### Socio-economic and demographic characteristics of respondents

Table 4 presents the socio-economic and demographic characteristics of the respondents, showing that the proportion of men (51.79%) was significantly higher than that of women in

**Table 4. Socio-economic and demographic characteristics.**

| ADH | 2 | 3 | 4 | 5 | 6 | All ADH | Statistic test |
|---|---|---|---|---|---|---|---|
| Gender (1 = man; 0 = woman) (%) | 52.19 | 50.91 | 51.67 | 51.12 | 51.79 | 51.61 | 67.36 *** |
| Class level reached (in years) | 4.67 (3.44) | 5.00 (2.44) | 6.15 (2.56) | 5.76 (3.00) | 5.96 (1.82) | 6.18 (2.80) | -3.53 *** |
| Number of active agricultural members | 12.03 (9.14) | 11.43 (5.75) | 7.56 (4.44) | 8.41 (5.49) | 8.10 (4.66) | 6.74 (5.69) | 2.78 *** |
| Access to credit (%) | 13.14 | 25.45 | 13.68 | 18.83 | 5.17 | 13.81 | 13.82 ** |
| Acreage (in Hectare) | 6.55 (4.96) | 8.81 (6.27) | 2.96 (1.55) | 1.83 (1.28) | 2.33 (1.03) | 3.8 (1.53) | 23.15 *** |

ADH = agricultural development hub.

any hub (p <0.01). With regard to the level of class reached, all producers had an average of seven (06) years. Significant differences were observed at the levels of the producers of ADH2 and ADH3, which totalled 5 years of class reached. On average, seven people made up a household in the total population. The significantly differences show that the household size of producers of ADH2 (12 people), ADH3 (11 people) is significantly higher than those of ADH4, ADH5, and ADH6 (8 people). Less than 14% of all producers had access to agricultural credit. The variation observed within the hubs showed that the level of access to credit for producers of ADH3 (25.45%) and ADH5 (18.83%) was significantly higher than that of ADH2 (13.14%), ADH4 (13.68%), and ADH6 (5.17%) (p <0.01). Likewise, the area planted with those of ADH2 (6.55 Ha) and ADH3 (8.81 Ha) was significantly larger than those of ADH6 (2.33 Ha), ADH5 (1.83 Ha), and ADH4 (2.33 Ha) (p <0.01).

## Assessment of the nutritional status of soil in ADHs

The data in table A1 in appendix present the average of the nutrient contents of the soils (soil parameter), and the degrees of intensity of the associated limitations, according to the PDAs (Fig 3). Following Fig 3, the parameters which induce severe limitations in the soils of producers of PDA2, PDA4, and PDA6 are mainly related to the phosphorus content, the sum of the cations, and the capacity cation exchange. On the other hand, at the level of PDA3 and PDA5, the limiting factors are linked to the cation exchange capacity for PDA3, and the phosphorus for PDA5.

## Results of econometric estimates of experimental choices

**Results of latent class logit.** 20,940 observations from the responses of 962 respondents for 1,047 plots were analysed using the LCL model as part of the technology choice

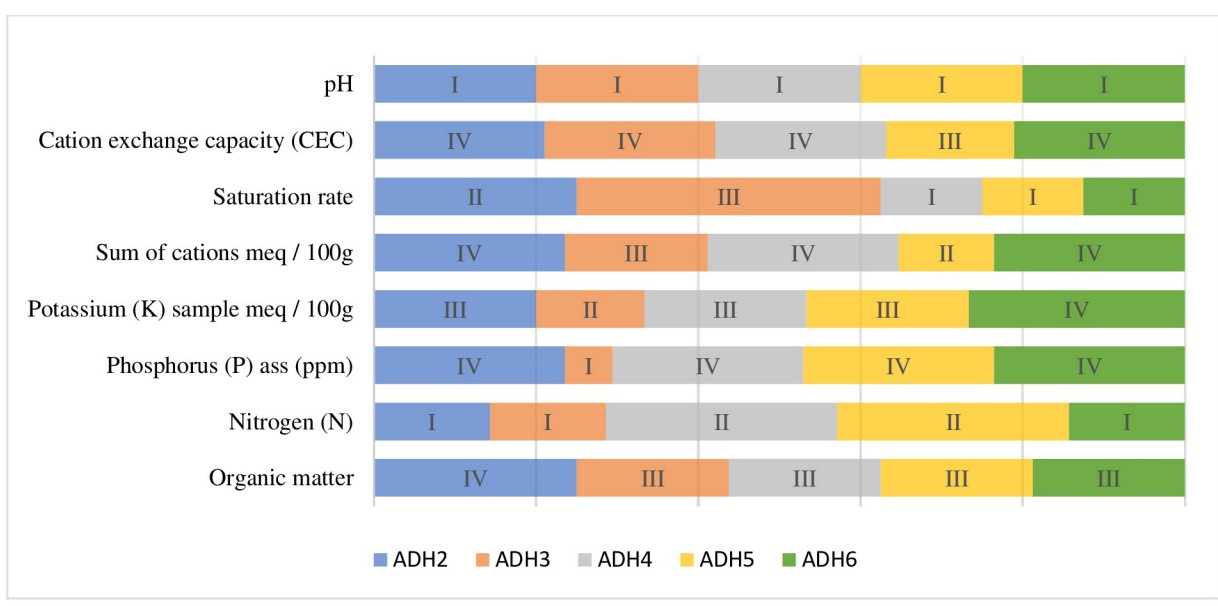

Degree I (Without limitations); Degree II (Weak limitations); Degree III (Average limitations); Degree IV (Severe Limitations);

Degree IV (Very severe limitations)

**Fig 3. Degree of intensity of associated limitations, according to ADHs.**

**Table 5. Calculation of Akaike (AIC), Bayesian (BIC), and Consistent (CAIC) information criteria.**

| Number of segments | Likelihood log | AIC | BIC | CAIC |
|---|---|---|---|---|
| 2 | −4658.60 | 9347.21 | 9398.16 | 9362.79 |
| 3 | −4607.61 | 9421.52 | 9165.34* | 9401.79 |
| 4 | −4551.67 | 9436.52 | 9318.90 | 9155.17* |
| 5 | −4545.79 | 9261.23* | 9349.90 | 9387.99 |
| 6 | −4530.58 | 9375.16 | 9169.59 | 9434.99 |

* indicates the lowest values of AIC, BIC, and CAIC.

experiment. Respondents' individual characteristics were assumed to affect their profile choices indirectly through their effect on class membership. The determination of the optimal number of classes requires the statistics reported in Table 5 to be used, mainly the Akaike, Bayesian, and Consistent information criteria (AIC, BIC, and CAIC, respectively), which must be minimal [41,53]. The results showed that AIC and BIC were respectively minimized to five and six segments, while the CAIC criterion was minimized to four segments. Five or six segments were already too high. In addition, the BIC criterion was the most reliable [26,41], and also allowed for a more robust explanation of the heterogeneity of preferences. Consequently, three segments were selected.

The results of the estimation of the logit model with three latent classes are presented in Table 6. As shown in Table 6, the first level presented the utility coefficients associated with the attributes of soil fertility technologies, while the second segment presented those of individual producer characteristics. The coefficients associated with the individual characteristics of the third segment producers were reduced as zero, i.e. the S3 segment was considered as a reference. The pseudo R2 of the model was 0.69, which suggests a very good fit. The predicted probability that a producer would belong to segments S1, S2, and S3 was 33%, 31%, and 36%, respectively. The distribution of the sample among the three classes finally appeared to be fairly balanced: 32.09% were assigned to S1, 30.95% to S2, and 36.96% to S3.

The estimated parameters of the utility function showed that the constants (*status quo*) specific to S1, S2, and S3, were highly and negatively significant in the model. Consequently, these classes of producers were predisposed to change their current situation (*status quo*) to adopt a new form of technology in view of their soil with a low fertility level (p <0.01).

Preferences were homogeneous across all segments for technology with a high restoration speed (subsequent rapid growth of crops and yields) and would favour the production of additional benefits (p <0.01), taking into account the positive sign of the coefficient associated with these variables. Also, all producers revealed a very strong preference for a technology that will be available at all times and easily accessible.

The three segments identified differ in their preference for attributes about cost, sustainability of soil conservation, frequency of plot maintenance after application of technology, and individual characteristics. In addition, Wald's test was able to show that the segments differed from each other mainly with respect to these attributes and individual characteristics because the value of the statistical test of the likelihood ratio estimated from the model (13,587.93) exceeded the critical value of 100.42 for a distribution at 60 degrees of freedom (p <0.01). Therefore, the null hypothesis of all parameters and interaction terms jointly equal to zero was rejected. The variable related to the level of education and soil pH had a non-significant coefficient for all classes. This implies that producers who have the same level of education and soil pH are randomly assigned to the three segments. This result justifies the relatively stable soil pH level at the level of the plots of most producers.

**Table 6. Estimation of the model 3 latent classes.**

| Attribute | Segment 1 | Segment 2 | Segment 3 |
|---|---|---|---|
| Status quo | −0.14 (0.08) ** | −3.10 (0.29) *** | −3.44 (0.32) *** |
| Cost | 0.04 (0.02) ** | −0.05 (0.01) ** | −0.34 (0.08) *** |
| Short restoration time | 1.17 (0.07) *** | 1.89 (0.12) *** | 3.83 (0.49) *** |
| Accessibility | 0.33 (0.06) *** | 0.65 (0.10) *** | 0.98 (0.10) *** |
| Possibility of obtaining additional benefits | 0.67 (0.12) *** | 0.99 (0.22) *** | 3.81 (0.49) *** |
| Long conservation time | 0.14 (0.09) | −1.68 (0.13) *** | 0.81 (0.17) *** |
| Maintenance frequency (regular) | 0.10 (0.07) * | 0.03 (0.01) ** | −0.11 (0.30) |
| Maintenance frequency (regular) × Cost | 0.28 (0.35) | −1.51 (0.98) *** | −0.82 (0.01) ** |
| Accessibility × Cost | 1.65 (1.19) ** | −0.14 (0.05) | −4.81 (2.66) ** |
| **Physicochemical characteristics of the soil** | | | |
| Organic Matter (OM) Rate | −0.14 (0.24) | 1.18 (0.40) *** | |
| N rate | 12.88 (4.31) *** | −1.85 (6.06) | |
| P rate | 0.01 (0.01)* | −0.64 (0.01)* | |
| K rate | −1.73 (0.03) ** | −0.19 (0.05)* | |
| Soil pH level | 0.25 (0.09) | −0.55 (0.12) | |
| Fertility level | −0.46 (0.01) ** | −0.28 (0.06) * | |
| Duration Fallow period | −0.05 (0.15) | −0.36 (0.08) ** | |
| **Individual characteristics** | | | |
| Gender (1 = man; 0 = woman) | 0.42 (0.30) | 0.62 (0.05) ** | |
| Formal education | −1.58 (0.35) | −0.53 (0.35) | |
| Number of active agricultural members | 0.67 (0.18) *** | 0.84 (0.26) *** | |
| Access to credit | 1.42 (0.36) *** | −0.18 (0.06) * | |
| Acreage | 0.15 (0.19) | 1.03 (0.20) *** | |
| ADH3 | −0.57 (1.08)* | 1.27 (0.83) | |
| ADH4 | −17.40 (27.38) | 6.79 (1.67) *** | |
| ADH5 | 2.13 (0.69) *** | 0.74 (0.84) * | |
| ADH6 | 19.19 (1.40) | 10.17 (8.68) *** | |
| Constant | −2.21 (0.71) *** | 0.73 (0.11) *** | |
| **Probability of belonging to each class** | **0.33** | **0.31** | **0.35** |
| **Number of individuals (%)** | **32.09** | **30.95** | **36.96** |
| Number of observations | 20,940 = 1047*5*4 | | |
| Number of respondents = 962; Number of plots = 1047 | | | |
| Likelihood log | −4322.48 | | |
| R2 | 0.69 | | |
| Test Wald Chi2(60) | 13587.93*** | | |

***, **,* mean, respectively, that the coefficients are significant at the 1%, 5%, and 10% threshold; numbers in parentheses represent standard errors.

Concerning class 1, the membership coefficients for this class of producers show that they were men from ADH2, ADH3, ADH4, ADH5, and ADH6 who sowed vast areas and practiced short fallow periods. In addition, given their ease of access to campaign credit and the high proportion of active farm members in their households, they revealed preferences for expensive technologies even if they require regular maintenance or monitoring of the plot after application.

The coefficients pertaining to the terms of interactions with maintenance cost and frequency, and accessibility were positive, indicating these producers' preferences for expensive accessible technologies even if they require regular maintenance. The coefficients of the

variables associated with fertility level and soil chemical components (organic matter content: OM; K) were negative, and those of the variables associated with nitrogen and phosphorus content were positive. This result indicates that the fertility status of their soil was practically low with high levels of N and P and low levels of OM and K.

Segment S2 of the model was significantly characterized by men from all ADHs who planted vast areas and practiced a short fallow period as in S1. The heterogeneity of preference between S1 and S2 appeared at the attribute levels specifying the duration of soil fertility conservation and cost. In addition, the variable related to the number of the active members and that of the attribute specifying the maintenance frequency were positive and showed that the probability of adopting technologies requiring high-frequency maintenance increases proportionally with the proportion of active agricultural members in the household of producers in S1 and S2. Conversely, the coefficient related to the term of interaction with the cost and the accessibility was negative, indicating these producers' preferences for cheaper technologies when they are available. Similarly, the coefficient associated with the interaction term, cost and maintenance frequency was negative. This result indicates that producers in this class were also oriented towards high-cost technologies that require less maintenance or work. The coefficients of the variables associated with fertility level and soil chemical components (N, P, and K) were negative, and those of the variables associated with OM levels were positive. This finding indicates that the fertility status of the soil was practically low with a low rate of N, P, K and a high rate of OM.

In relation to the individual producers' characteristics in segment S3, which is considered as a reference segment, the parameter coefficients may be interpreted as regards the coefficients of the other two segments provided that these coefficients are all of the same sign and significance [26]. However, the S3 class was significantly characterized by men and women from ADH2, ADH3, ADH4, and ADH6, who practiced fallowing over a long period of time with preferences for less-expensive technologies. The fertility level of their soil was relatively low with a very high K level. The sign of the utility coefficients of the segment variables (S3) revealed that producers of this class shared almost the same preferences with those in S2. Indeed, preferences were heterogeneous with regard to the attribute relating to the duration of conservation and maintenance frequency after the application of technology. All producers in S3 revealed a determining position for all attributes except for the one with regard to maintenance frequency, which was not significant. This result indicates that these producers did not also attach particular importance to the frequency of maintenance after the technology was applied. Nevertheless, the sign of the coefficient of the variable shows that they had a preference for technologies that promote soil fertility over a long period of time and require less maintenance.

**Results of endogenous attribute attendance model.** The results from the EAA model are presented in Table 7 with the probabilities of ANA. In model 1, the attributes specifying the possibility of obtaining an additional benefits had the lowest significant ANA probabilities, which indicates that the probability that the attribute linked to the possibility of obtaining an additional benefit, is ignored in a choice situation is 13%. Lower probabilities were obtained for attributes pertaining to restoration speed (7%). The probabilities regarding to cost and accessibility were not significant, indicating that these attributes do not explain the probability of ANA. The attribute most often ignored was sustainability (86%) followed by maintenance frequency (52%), which was then excluded from model 2.

Model 2 presents estimates jointly with model 1, according to the attributes excluded from the choice in model 1. As a result, model 2 involves only the attributes with the lowest probability of rejection in model 1. In model 2, the attributes specifying the sustainability and the frequency of maintenance were excluded. These variables were jointly ignored at 57% when they were in competition with the other attributes (cost, short restoration time, accessibility,

**Table 7. Results of endogenous attribute attendance model.**

| Attribute | Model 1 | | Model 2 | |
|---|---|---|---|---|
| | Coefficient | *p* ANA | Coefficient | *p* ANA |
| Status quo | −3.28 (0.40) *** | 0.69 (0.03) *** | −2.74 (0.2) *** | 0.66 (0.03) *** |
| Cost | −0.10 (0.01) *** | 0.01 (0.03) | −0.09 (0.01) *** | |
| Restoration time | 2.13 (0.09) *** | 0.07 (0.01) *** | 2.11 (0.10) *** | 0.08 (0.02) *** |
| Accessibility | 0.66 (0.13) *** | 0.10 (0.16) | 0.56 (0.03) *** | 0.02 (0.01) |
| Possibility of obtaining additional benefits | 1.92 (0.20) *** | 0.13 (0.05) ** | 1.77 (0.21) *** | 0.13 (0.05) ** |
| Soil fertility retention time | 2.95 (0.32) *** | 0.86 (0.02) *** | 1.19 (0.04) *** | |
| Frequency of maintenance of the plot | −0.15 (0.06) ** | 0.52 (0.13) ** | −0.086 (0.05) | |
| *p* excluded attribute | | | | 0.57 (0.10) *** |
| Number of observations | 20940 | | 20940 | |
| Likelihood log | −4675.32 | | −4710.56 | |
| Wald Chi2(8) | 806.48 *** | | 1109.68*** | |
| AIC | 9376.65 | | 9445.13 | |
| BIC | 9479.99 | | 9540.52 | |

***, **,* mean, respectively, that the coefficients are significant at the 1%, 5% and 10% threshold; *p*: probability of ANA.

and possibility of obtaining additional benefits). These latter attributes tend to have a low probability of rejection (<13%) when confronted together in a choice process. This indicates that these attributes are likely to play an essential role in the decision-making process. The likelihood of rejecting the status quo in both models was high, implying that respondents pay more attention to alternative situations during choices. Therefore, they are interested in new recommendations regarding the management of soil fertility.

**Estimate of the willingness to adopt.** The parameters estimated from the LCL model were used to calculate the WTA for each attribute at the level of each segment (Fig 4) from the

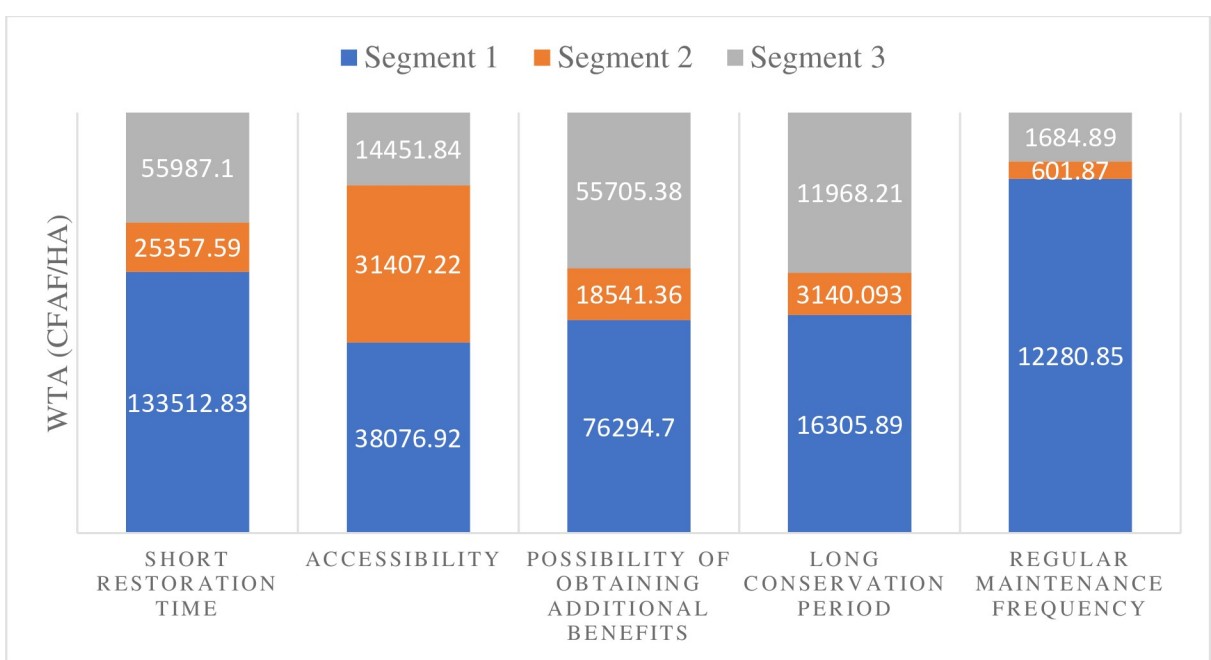

**Fig 4. Estimate of the willingness-to-adopt from the three latent classes model.**

derivative of the cost-related monetary attribute. This makes it possible to understand the motivation of the different classes of stakeholders and quantify their levels of preference. The results presented in Fig 4 show that the producers of segment S1 (CFAF 133,512.83 or 220.05 dollars US, at a fixed exchange rate of 1 dollar US = CFAF 606,73), S2 (CFAF 33,357.59), and S3 (CFAF 55,987.00) were more interested in the attribute linked to the time to restore soil fertility compared with other attributes.

With the exception of from the restoration time, attributes linked to the possibility of obtaining additional benefits and accessibility were also valued. Represented in order of importance, the attributes regarding to short restoration time, possibility of obtaining additional benefits, and accessibility were more popular with producers of S1 and S3. Those of S2 prioritized, in order of importance, short restoration time, accessibility, and possibility of obtaining additional benefits. Controlling for the attribute linked to long shelf life, it was found that this attribute was important to those in S1 (CFAF 16,305.89) and S3 (CFAF 11,968.21) but not those in S2. This result indicates that those of S2 did not attach any particular importance to the duration of restoration of soil fertility. By considering the attribute specifying the frequency of regular maintenance, it appears that the producers of S1 and S2 considered this attribute positive unlike those of S3.

## Discussion

The producers were predisposed to change their current situation and follow other recommendations. The main attributes prioritized during the choices relate to cost, short recovery time, accessibility, and the possibility of obtaining additional benefits. The assessment of the heterogeneity of producer preferences made it possible to identify three classes of producers, appearing in different proportions (S1: 32.09%; S2: 30.95%; and S3: 36.96% in S3). The heterogeneity of preferences was observed at the level of attributes concerning to cost, duration of soil conservation, and frequency of maintenance of the plot after application of the technology. On the other hand, preferences were homogeneous across all segments, with respect to technology whose speed of recovery is fast (subsequently rapid growth of crops and yields), and would be able obtain additional benefits. In addition, technologies favouring the obtaining of additional benefits promote not only soil fertility, but also contribute to the income and food security of farmers through the provision of edible or marketable products. These results confirm those of Yabi et al. [45] who found that the technology about to the use of Mucuna, Aeschinomene Hytrix is less used because it does not facilitate the obtaining of an edible food.

All the producers show a very strong preference for a technology that will be available at all times, and easily accessible. This result agrees with that of Assogba et al. [46], and Maliki [47] which shows that supply difficulties and lack of funding limit the use of technologies. Katengeza [54] finds that the unavailability of the technology subsequently leads to non-compliance with recommended doses. Likewise, Adekambi et al. [55] assert that the adopters of aqueous botanical extracts abandoned this practice after having experienced it themselves at least once on their respective plots because of the problems of unavailability of the leaves of these extracts.

The heterogeneity of preferences shows that class S1 brings together men from ADH2, ADH3, ADH4, ADH5, and ADH6, who cultivate large areas, and who belong to a group. In addition, given their ease of accessing seasonal credit, and the high proportion of active agricultural members in their household, they reveal preferences for expensive technologies even if they require regular maintenance or control of the plot. after application. This result can also be explained by the fact that they consider the price to be a good quality indicator. Also, credit for them can be an for the poor to invest. In addition, the income situation of producers leaves

little room for self-financing. IFS [56], and Yabi et al. [45] show that agricultural financing is important for the viability of the soil fertility management action plan in Benin. Also, they had an attraction for expensive accessible technologies, even if they require regular maintenance. Given the high cost of the technology, they want to perform regular maintenance in order to benefit from its economic performance over several production campaigns. These observations confirm the research results of Maliki [47], and of Baco et al. [57], who show that technologies that require regular field control also promote weed control, which may make the soil fertility conservation technology in use more effective. The fertility status of their soil was practically low, with high nitrogen and phosphorus content, and low in organic matter and potassium. In view of their preference, they were named "segment of men who are major producers of ADH 2, 3, 4, 5, and 6, given the size of their large area, having access to credit with infertile soils (especially with low OM and K rate), users of a costly practice in terms of purchase and use (maintenance), easily accessible, favouring acceleration, sustainability of soil fertility, and obtaining additional benefits. Indeed, the characteristics of the technology desired by these producers were similar to the use of crop residues. As a result, the use of crop residues would promote the delivery of nutrients rich in organic elements to the soils, and mineral fertilizers promote the delivery of the mineral elements. This result agrees with that of RAMR [58] who shows that the practice concerning the use of crop residues cannot be done over a large area, because the storage of residues requires a very large labor of plot.

Segment (S2) was characterized by less educated producers of ADH3, ADH4, ADH5, and ADH6 who practiced a short fallow like those in S1. Given the difficulties in accessing seasonal credit, they reveal preferences for technologies that promote soil conservation over a short period (one production campaign), and less expensive. This result supports the findings of Abbasi et al. [59,60] who show that the real economic activity grows when the price of electricity decreases and the electricity demand rises.

Poverty is, after ignorance, one of the factors limiting the adoption of technological approaches, and investment in the regeneration of land fertility. Under these conditions, the granting of well-studied seasonal credits is more appropriate [3,57].

In addition, the probability of adoption of technologies requiring a high maintenance frequency increases proportionally with the proportion of members of agricultural labor in the household of producers in S1 and S2. The preferences of the producers in the S2 and S3 segment were also oriented towards technologies that are expensive, but require less maintenance. They believe that new technologies should help simplify work. The position of this segment of producers is particularly evident with regard to technologies requiring regular maintenance, involving the use of important production factors that require significant costs. For the most part, in order to minimize costs and achieve profit margins, they prefer less expensive technologies, which do not incur significant production costs. The fertility status of their soil was practically low with a low level of N, P, K, and a high organic content. They were qualified as a segment of producers who do not have access to credit with infertile soils (low rate of N; P; K and high rate of OM), users of a practice that is less expensive in terms of purchase, easily accessible and requiring regular control of the field, obtaining additional benefits favouring the acceleration and conservation of soil fertility over a production campaign. Indeed, the characteristics of the technology desired by these producers were similar to those of the use of mineral fertilizers (potash fertilizers) given the low level of N; P; K, and a high level of organic matter. These results are similar to the findings of Krah et al. [61] in level of smallholders of Malawi.

Class S3 has been characterized by men, and women from ADH2, ADH3, ADH4, and ADH6, who are not in groupings, and who have an attraction for less expensive technologies. The fertility of their soil was marked only by a high potassium element content. They were

attracted to technologies that favour soil fertility over a long period of time, and which require less maintenance.

The position of producers in this segment is particularly evident in view of the small size of the active members of the household. Indeed, the characteristics of the technology preferred by these producers were similar to those of the practice of crop rotation. This technique is preferred in order to regenerate, and guarantee the conservation of the soil structure over a long period. Sanginga and Woomer [62], reveal that rotations integrating legumes directly contribute to the constitution of soil organic matter, which plays multiple functions in improving the physicochemical and biological characteristics of soils. They were characterized by men and women of ADH2, ADH3, ADH4, and ADH6 having soils rich in potassium element, users of less expensive practice in terms of purchase and use (maintenance), favouring acceleration and sustainability of soil fertility, and obtaining additional benefits.

## Conclusion

The main concern of small farmers in Africa remains the management and conservation of soil fertility from one season to another to ensure food supplies and improve well-being. This work is important to highlight the serious fertility problems in Benin. This precise research study offers an original method to determine and examine the heterogeneity of preferences by identifying different segments of producers with particular preferences for the attributes of soil fertility management technologies at the ADH level. The results of the experimental choices of technology profiles for the plots show that the majority of producers are inclined to opt for novel recommendations given the state of their relatively infertile soil. All producers, regardless of gender or ADH, tended to prioritize accessible technologies, favouring improvements in soil fertility and the ability to produce additional benefits. These attributes, added to that relating to cost, tended to have a low probability of rejection during the selection process. Preferences were heterogeneous for attributes specifying cost, sustainability of soil conservation, frequency of plot maintenance after technology was applied, and individual characteristics. However, three classes of producers were identified. Class S1 was referred to as: segment of large male producers of ADH2, ADH3, ADH4, ADH5, and ADH6, with access to credit and infertile soils (especially low levels of OM and K) and users of practices that are expensive in terms of purchase and use (maintenance), easily accessible, promote the improvement and sustainability of soil fertility, and allow for producing additional benefits. Indeed, the typical technology profile desired by S1 women was similar to that for the use of crop residues. Thus, except fr mineral fertilizers, this practice was the most preferred by this class of producers (S1) unlike those of the other segments (S2 and S3) given the relatively low fertility level of their soil with high levels of N and P and low levels of OM and K. In addition, the rotation technique was preferred to guarantee the conservation of soil structure over a long period of time. Those in class S2 were qualified as large producers who do not have access to credit but do have infertile soils (low rate of N, P, and K and high rate of OM) and users of practices that are less costly in terms of purchase, easily accessible, and require regular field monitoring, and allow for the production of additional benefits that promote the improvement and conservation of soil fertility over a production campaign. In addition, the characteristics of the technology preferred by these producers were similar to those for mineral fertilizer use. This technology was preferred given the low level of soil fertility with a low rate of N, P, and K and a high rate of OM. The characteristics of the technology desired by those in class S3 were similar to those of the crop rotation practice. From their assessment, we can see that they were characterized as men and women from ADH2, ADH3, ADH4, and ADH6 who have less fertile soils and are users of practices that are less expensive in terms of purchase and use (maintenance), easily accessible,

promote the acceleration and sustainability of soil fertility, and allow for the production of additional benefits. In short, these results show that all producers are not driven by the same expectations. The critical contribution of this research study, from a theoretical viewpoint, is the analysis of the respondents of ADHs, and their understanding of the soil fertility technological profiles, which, in turn, could help business firms to attain sustainable performance.

## Policy recommendations

The results of this research will enable local decision-makers to measure possible interest in spatially discriminating soil fertility management measures by ADH. The definition of these measures at the technical level must be based on the frequent support of extension and technical agents towards producers, the technical performance of technologies, and the installation of school plots. At the economic level, strategies aimed at financial support (e.g. subsidies and grants) and facilitating access to agricultural credit must be implemented. The institutional and political measures relate to the ease of access in villages through the construction of infrastructure, definitions of accompanying measures, and access and availability of technologies. Promoting the advantages of the technologies sought by ADH through channels responsible for the dissemination of information or awareness and popularization campaigns would encourage the implementation of technologies. Furthermore, the development of a sectoral policy that takes into account the monitoring of the specificity of each ADH would also encourage the implementation of technologies. The objective is to create an adequate environment in which farmers can acquire the technologies they need to meet their demands for sustainable supply and support sectors. Moreover, it will be necessary to strengthen the cooperation between the public and private sectors for developing technologies to encourage the use of technologies by end users and facilitate access to markets. Technologies should be applied by users with sustainability criteria for rational biomass management. An interesting perspective would be to explore the preferences of producers of other ADHs (1 and 7) that are not associated with the SAPEP intervention areas.

## Supporting information

**S1 Fig. Study zone.**
(DOCX)

**S2 Fig. Example of a set of cards proposed during the interview.**
(DOCX)

**S3 Fig. Degree of intensity of associated limitations, according to ADHs.**
(DOCX)

**S4 Fig. Estimate of the willingness-to-adopt from the three latent classes.**
(DOCX)

**S1 Table. Criteria for assessing the degrees of limitation of soil chemical parameters.**
(DOCX)

**S2 Table. Attributes and associated attribute levels.**
(DOCX)

**S3 Table. Variables used in econometric models.**
(DOCX)

**S4 Table. Socio-economic and demographic characteristics.**
(DOCX)

**S5 Table. Calculation of Akaike (AIC), Bayesian (BIC), and Consistent (CAIC) information criteria.**
(DOCX)

**S6 Table. Estimation of the model 3 latent classes.**
(DOCX)

**S7 Table. Results of endogenous attribute attendance model.**
(DOCX)

**S1 Appendix. Average nutrient content of soils, and the degree of intensity of associated limitations, according to ADHs.**
(DOCX)

**S1 File. Survey guide.**
(DOCX)

## Acknowledgments

We acknowledge the financial assistance from of Smallholder Agricultural Productivity Enhancement Program in Sub-Saharan Africa (SAPEP). The authors would also like to thank Professor Afio Zannou for earlier works. We further acknowledge the help from the research group of International Center of Research and Training in Social Science (CIRFOSS), and those of National Agricultural Research Institute of Benin (INRAB) for proof-reading the thesis. We also thank Elsevier Language Editing Services.

## Author Contributions

**Conceptualization:** Segla Roch Cedrique Zossou, Patrice Ygue Adegbola, Brice Tiburce Oussou, Gustave Dagbenonbakin, Roch Mongbo.

**Data curation:** Segla Roch Cedrique Zossou, Brice Tiburce Oussou, Gustave Dagbenonbakin, Roch Mongbo.

**Formal analysis:** Segla Roch Cedrique Zossou, Patrice Ygue Adegbola, Brice Tiburce Oussou, Roch Mongbo.

**Funding acquisition:** Patrice Ygue Adegbola.

**Investigation:** Segla Roch Cedrique Zossou, Brice Tiburce Oussou.

**Methodology:** Segla Roch Cedrique Zossou, Patrice Ygue Adegbola, Brice Tiburce Oussou, Gustave Dagbenonbakin, Roch Mongbo.

**Project administration:** Segla Roch Cedrique Zossou, Patrice Ygue Adegbola, Brice Tiburce Oussou, Gustave Dagbenonbakin, Roch Mongbo.

**Resources:** Segla Roch Cedrique Zossou, Patrice Ygue Adegbola, Brice Tiburce Oussou, Gustave Dagbenonbakin, Roch Mongbo.

**Software:** Segla Roch Cedrique Zossou, Patrice Ygue Adegbola, Brice Tiburce Oussou, Gustave Dagbenonbakin, Roch Mongbo.

**Supervision:** Segla Roch Cedrique Zossou, Patrice Ygue Adegbola, Brice Tiburce Oussou, Gustave Dagbenonbakin, Roch Mongbo.

**Validation:** Segla Roch Cedrique Zossou, Patrice Ygue Adegbola, Brice Tiburce Oussou, Gustave Dagbenonbakin, Roch Mongbo.

**Visualization:** Segla Roch Cedrique Zossou, Patrice Ygue Adegbola, Brice Tiburce Oussou, Gustave Dagbenonbakin, Roch Mongbo.

**Writing – original draft:** Segla Roch Cedrique Zossou, Patrice Ygue Adegbola, Brice Tiburce Oussou, Gustave Dagbenonbakin, Roch Mongbo.

**Writing – review & editing:** Segla Roch Cedrique Zossou, Patrice Ygue Adegbola, Brice Tiburce Oussou, Gustave Dagbenonbakin, Roch Mongbo.

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
