## [Decision Letter · Decision Letter 0]

3 May 2021

PONE-D-21-07803

Modelling Smallholder Farmers' Preferences for Soil Fertility Management Technologies in Benin: A Stated Preference Approach

PLOS ONE

Dear Dr. Segla Roch Cedrique Zossou,

Thank you for submitting your manuscript to PLOS ONE. After careful consideration, we feel that it has merit but does not fully meet PLOS ONE’s publication criteria as it currently stands. Therefore, we invite you to submit a revised version of the manuscript that addresses the points raised during the review process.

We look forward to receiving your revised manuscript.

Kind regards,

Carlos Alberto Zúniga-González, Ph.D

Academic Editor

PLOS ONE

Additional Editor Comments:

No comment

Journal Requirements:

2. Please include additional information regarding the survey or interview guide used in the study and ensure that you have provided sufficient details that others could replicate the analyses. For instance, if you developed a questionnaire or interview guide as part of this study and it is not under a copyright more restrictive than CC-BY, please include a copy, in both the original language and English, as Supporting Information.

4. We note that Figure 2 includes an image of aparticipant in the study. 

Reviewers' comments:

Reviewer's Responses to Questions

**Comments to the Author**

1. Is the manuscript technically sound, and do the data support the conclusions?

Reviewer #1: Yes

Reviewer #2: Yes

2. Has the statistical analysis been performed appropriately and rigorously? 

Reviewer #1: Yes

Reviewer #2: Yes

3. Have the authors made all data underlying the findings in their manuscript fully available?

Reviewer #1: Yes

Reviewer #2: Yes

4. Is the manuscript presented in an intelligible fashion and written in standard English?

Reviewer #1: Yes

Reviewer #2: Yes

5. Review Comments to the Author

Reviewer #1: I found this study informative, which presents a new idea entitled, " Modelling Smallholder Farmers' Preferences for Soil Fertility Management Technologies in Benin: A Stated Preference Approach."

This article states that the decline of soil fertility is a major constraint which results in lower levels of crop productivity, and agricultural development and food security in Sub-Saharan Africa. This study analyses the most interesting technological profiles to offer to each category of producer in Benin agricultural development hubs (ADHs) using the stated preference method, more precisely, the improved choice experiment method. The investigation focused on 1047 sampled plots from 962 randomly selected producers in villages of the Smallholder Agricultural Productivity Enhancement Program in Sub-Saharan Africa of the ADHs.

Abstract and Introduction improvement:

I am glad to assess this informative study. In my opinion, I have some guidelines for the authors to enhance the study quality before endorsing it for publication. As the Abstract is the main door or "FACE" of the manuscript, it should briefly present high-quality English with new information. I am recommending the authors of this study to expand Abstract, as it is too short. The Abstract should be around 250 words. I have suggested some studies to check the abstracts and improve yours and cite them in the introduction and build your study objectives like these studies.

Hussain, T., Abbas, J., Wei, Z., & Nurunnabi, M. (2019). The Effect of Sustainable Urban Planning and Slum Disamenity on The Value of Neighboring Residential Property: Application of The Hedonic Pricing Model in Rent Price Appraisal. Sustainability, 11(4). doi:10.3390/su11041144

Abbas, J., Raza, S., Nurunnabi, M., Minai, M. S., & Bano, S. (2019). The Impact of Entrepreneurial Business Networks on Firms’ Performance Through a Mediating Role of Dynamic Capabilities. Sustainability, 11(11). https://doi.org/10.3390/su11113006

Hussain, T., Abbas, J., Wei, Z., Ahmad, S., Xuehao, B., & Gaoli, Z. (2021). Impact of Urban Village Disamenity on Neighboring Residential Properties: Empirical Evidence from Nanjing through Hedonic Pricing Model Appraisal. Journal of Urban Planning and Development, 147(1), 04020055. https://doi.org/10.1061/(asce)up.1943-5444.0000645

Literature section

It presents a good summary of the literature. I suggest authors add the literature as recommended below to improve the manuscript. Overall, the authors have creatively linked variables. It reflects an innovative model of the study. I am pleased to read this article. However, I have some suggestions for the authors to enhance the quality of the literature section. The authors can add few lines about technological innovations and environmental responsibility practices. Please see the suggested studies and cite them to enhance the literature section.

Abbas, J., Zhang, Q., Hussain, I., Akram, S., Afaq, A., & Shad, M. A. (2020). Sustainable Innovation in Small Medium Enterprises: The Impact of Knowledge Management on Organizational Innovation through a Mediation Analysis by Using SEM Approach. Sustainability, 12(6). https://doi.org/10.3390/su12062407

Methods and Results

The results section of the paper presents a good view of the study. This work presents a notable investigation on a selected topic. I suggest including some graphical presentations to improve the quality of this study. Please see the proposed studies and see the graphical representation. Improve your work like these studies and cite them in this section.

Abbasi, K. R., Abbas, J., & Tufail, M. (2021). Revisiting electricity consumption, price, and real GDP: A modified sectoral level analysis from Pakistan. Energy Policy, 149, 112087. doi:10.1016/j.enpol.2020.112087

Abbas, J., Aman, J., Nurunnabi, M., & Bano, S. (2019). The Impact of Social Media on Learning Behavior for Sustainable Education: Evidence of Students from Selected Universities in Pakistan. Sustainability, 11(6). https://doi.org/10.3390/su11061683

Abbasi, K. R., Hussain, K., Abbas, J., Adedoyin, F. F., Shaikh, P. A., Yousaf, H., & Muhammad, F. (2021). Analyzing the role of industrial sector's electricity consumption, prices, and GDP: A modified empirical evidence from Pakistan [J]. AIMS Energy, 9(1), 29-49. doi:10.3934/energy.2021003

Abbas, J., Mahmood, S., Ali, H., Ali Raza, M., Ali, G., Aman, J., . . . Nurunnabi, M. (2019). The Effects of Corporate Social Responsibility Practices and Environmental Factors through a Moderating Role of Social Media Marketing on Sustainable Performance of Business Firms. Sustainability, 11(12), 3434.

Conclusion

I suggest you make a separate heading of the conclusion and do not mix it with implications.

Policy Recommendations

I again recommend you to make a separate heading of the Policy Recommendations.

The conclusion section is acceptable. Overall, this presents a good piece of research work. I recommend that authors do a little more work and revise this article accordingly. I suggest the authors check English quality and fix some weak sentences. If you have already taken English editing service, ask them to recheck the quality to meet scientific merit for publication. I endorse this manuscript for publication after minor corrections, as suggested.

Reviewer #2: I am glad to review and assess this interesting article, entitled, Modeling Smallholder Farmers' Preferences for Soil Fertility Management Technologies in Benin: A Stated Preference Approach. This study analyses the most interesting technological profiles to offer to each category of producer in Benin agricultural development hubs (ADHs) using the stated preference method, more precisely, the improved choice experiment method. The organization of this article is good and satisfactory. The Introduction section and methodology portions are adequate. I suggest the authors improve the Materials and Methods section by adding some latest articles' citations to enhance the work quality and also concise this part. Also, Improve the Conclusion part as well.

Overall, the manuscript is a good piece of work. I recommend that authors do a little more work and add the latest literature to support the study, as suggested. The English level is good and smooth, e.g., the language standard, specifically the grammar, of sufficient quality to meet scientific merit for publication. I accept this manuscript after minor revision, as I have recommended.

6. PLOS authors have the option to publish the peer review history of their article (what does this mean?). If published, this will include your full peer review and any attached files.

Reviewer #1: No

Reviewer #2: No

---

## [Author Response · Author response to Decision Letter 0]

2 Jun 2021

Reply to reviewers’ comments on PONE-D-21-07803R1

“Modelling Smallholder Farmers' Preferences for Soil Fertility Management Technologies in Benin: A Stated Preference Approach.”

Response to academic editor

Reply to specific comments:

1. We note the following text is only included in the Tracked Changes document: « The study was carried out on private land. We confirm that, the owner of the land gave permission to conduct the study on this site, and no specific permissions were required for these locations/activities. The agents of the research centers, the town hall, the village chiefs, and the resource persons facilitated the interviews with the producers in the villages."

Please confirm and include the above permissions text in the Methods section of your manuscript.

- Response 1: The correction has been made as suggested. We confirm and include the following text concerning the permissions text in the Methods section of the manuscript. " The study was carried out on private land. We confirm that, the owner of the land gave permission to conduct the study on this site, and no specific permissions were required for these locations/activities. The agents of the research centers, the town hall, the village chiefs, and the resource persons facilitated the interviews with the producers in the villages."

\f

Reply to reviewer 1:

First, we want to thank te reviewer for the time spent going through this paper for comments. We are convinced that these comments have substantially improved the paper.

Abstract and Introduction improvement:

I am glad to assess this informative study. In my opinion, I have some guidelines for the authors to enhance the study quality before endorsing it for publication. As the Abstract is the main door or "FACE" of the manuscript, it should briefly present high-quality English with new information. I am recommending the authors of this study to expand Abstract, as it is too short. The Abstract should be around 250 words. I have suggested some studies to check the abstracts and improve yours and cite them in the introduction and build your study objectives like these studies.

Hussain, T., Abbas, J., Wei, Z., & Nurunnabi, M. (2019). The Effect of Sustainable Urban Planning and Slum Disamenity on The Value of Neighboring Residential Property: Application of The Hedonic Pricing Model in Rent Price Appraisal. Sustainability, 11(4). doi:10.3390/su11041144

Abbas, J., Raza, S., Nurunnabi, M., Minai, M. S., & Bano, S. (2019). The Impact of Entrepreneurial Business Networks on Firms’ Performance Through a Mediating Role of Dynamic Capabilities. Sustainability, 11(11). https://doi.org/10.3390/su11113006

Hussain, T., Abbas, J., Wei, Z., Ahmad, S., Xuehao, B., & Gaoli, Z. (2021). Impact of Urban Village Disamenity on Neighboring Residential Properties: Empirical Evidence from Nanjing through Hedonic Pricing Model Appraisal. Journal of Urban Planning and Development, 147(1), 04020055. https://doi.org/10.1061/(asce)up.1943-5444.0000645

- Response : We thank the reviewer for this proposal. We expand the Abstract as suggested. Some studies suggested by the reviewer have been exploited and cited in order to improve the abstract and the introduction and build our study objectives.

Literature section

It presents a good summary of the literature. I suggest authors add the literature as recommended below to improve the manuscript. Overall, the authors have creatively linked variables. It reflects an innovative model of the study. I am pleased to read this article. However, I have some suggestions for the authors to enhance the quality of the literature section. The authors can add few lines about technological innovations and environmental responsibility practices. Please see the suggested studies and cite them to enhance the literature section.

Abbas, J., Zhang, Q., Hussain, I., Akram, S., Afaq, A., & Shad, M. A. (2020). Sustainable Innovation in Small Medium Enterprises: The Impact of Knowledge Management on Organizational Innovation through a Mediation Analysis by Using SEM Approach. Sustainability, 12(6). https://doi.org/10.3390/su12062407

- Response : The correction has been made as suggested by the reviewer. We have added the literature as recommended 

The results section of the paper presents a good view of the study. This work presents a notable investigation on a selected topic. I suggest including some graphical presentations to improve the quality of this study. Please see the proposed studies and see the graphical representation. Improve your work like these studies and cite them in this section.

Abbasi, K. R., Abbas, J., & Tufail, M. (2021). Revisiting electricity consumption, price, and real GDP: A modified sectoral level analysis from Pakistan. Energy Policy, 149, 112087. doi:10.1016/j.enpol.2020.112087

Abbas, J., Aman, J., Nurunnabi, M., & Bano, S. (2019). The Impact of Social Media on Learning Behavior for Sustainable Education: Evidence of Students from Selected Universities in Pakistan. Sustainability, 11(6). https://doi.org/10.3390/su11061683

Abbasi, K. R., Hussain, K., Abbas, J., Adedoyin, F. F., Shaikh, P. A., Yousaf, H., & Muhammad, F. (2021). Analyzing the role of industrial sector's electricity consumption, prices, and GDP: A modified empirical evidence from Pakistan [J]. AIMS Energy, 9(1), 29-49. doi:10.3934/energy.2021003

Abbas, J., Mahmood, S., Ali, H., Ali Raza, M., Ali, G., Aman, J., . . . Nurunnabi, M. (2019). The Effects of Corporate Social Responsibility Practices and Environmental Factors through a Moderating Role of Social Media Marketing on Sustainable Performance of Business Firms. Sustainability, 11(12), 3434.

- Response : The observation are very relevant and has been taken into account as suggested by the reviewer. Some graphical presentations are now added to improve the quality of this study. Some studies suggested by the reviewer have been exploited and cited in the manuscript

Conclusion

I suggest you make a separate heading of the conclusion and do not mix it with implications.

- Response : The correction has been made as suggested by the reviewer.

Policy Recommendations

I again recommend you to make a separate heading of the Policy Recommendations.

The conclusion section is acceptable. Overall, this presents a good piece of research work. I recommend that authors do a little more work and revise this article accordingly. I suggest the authors check English quality and fix some weak sentences. If you have already taken English editing service, ask them to recheck the quality to meet scientific merit for publication. I endorse this manuscript for publication after minor corrections, as suggested.

- Response : The correction is done as suggested by the reviewer. We have made a separate heading of the Policy Recommendations. We have checked English quality and fix some weak sentences with English editing service

\f

Response to Reviewer #2 Comments

I am glad to review and assess this interesting article, entitled, Modeling Smallholder Farmers' Preferences for Soil Fertility Management Technologies in Benin: A Stated Preference Approach. This study analyses the most interesting technological profiles to offer to each category of producer in Benin agricultural development hubs (ADHs) using the stated preference method, more precisely, the improved choice experiment method. The organization of this article is good and satisfactory. The Introduction section and methodology portions are adequate. I suggest the authors improve the Materials and Methods section by adding some latest articles' citations to enhance the work quality and also concise this part. Also, Improve the Conclusion part as well. Overall, the manuscript is a good piece of work. I recommend that authors do a little more work and add the latest literature to support the study, as suggested. The English level is good and smooth, e.g., the language standard, specifically the grammar, of sufficient quality to meet scientific merit for publication. I accept this manuscript after minor revision, as I have recommended.

Reply to general comments:

- First, we express our gratitude to the reviewer for devoting his time to this paper for the comments and for providing many relevant articles to the topic addressed in this paper. We are convinced that these comments and the review of suggested articles have substantially improved the paper. The correction has been made as suggested. We have improved the Materials and Methods, and Conclusion section, by adding some latest articles' citations to enhance the work quality and also concise this part.

---

## [Decision Letter · Decision Letter 1]

7 Jun 2021

Modelling Smallholder Farmers' Preferences for Soil Fertility Management Technologies in Benin: A Stated Preference Approach

PONE-D-21-07803R1

Dear Dr. Segla Roch Cedrique Zossou,

We’re pleased to inform you that your manuscript has been judged scientifically suitable for publication and will be formally accepted for publication once it meets all outstanding technical requirements.

Kind regards,

Carlos Alberto Zúniga-González, Ph.D

Academic Editor

PLOS ONE

Additional Editor Comments (optional):

Reviewers' comments:

Reviewer's Responses to Questions

**Comments to the Author**

1. If the authors have adequately addressed your comments raised in a previous round of review and you feel that this manuscript is now acceptable for publication, you may indicate that here to bypass the “Comments to the Author” section, enter your conflict of interest statement in the “Confidential to Editor” section, and submit your "Accept" recommendation.

Reviewer #1: All comments have been addressed

Reviewer #2: All comments have been addressed

2. Is the manuscript technically sound, and do the data support the conclusions?

Reviewer #1: Yes

Reviewer #2: Yes

3. Has the statistical analysis been performed appropriately and rigorously? 

Reviewer #1: Yes

Reviewer #2: Yes

4. Have the authors made all data underlying the findings in their manuscript fully available?

Reviewer #1: Yes

Reviewer #2: Yes

5. Is the manuscript presented in an intelligible fashion and written in standard English?

Reviewer #1: Yes

Reviewer #2: Yes

6. Review Comments to the Author

Reviewer #1: I am satisfied to evaluate the revised manuscript. I have found the revised version of this study effective and satisfactory. The authors have made a good attempt and answered all my points to improve the quality of this study. I feel happy to avail myself of the opportunity to evaluate this informative study.

In my evaluation, this version of the article entitled, "Modelling Smallholder Farmers' Preferences for Soil Fertility Management Technologies in Benin: A Stated Preference Approach" has reach merit for publication. I believe that the authors have made an excellent revision to reach scientific merit for the publication of this study. The article is well structured, and the methodology is appropriate, well applied, and discussed. I accept and endorse this revised article in the current format, as the authors have made a satisfactory revision to achieve scientific merit for publication. Have a smooth publication procedure. Good Luck!

Reviewer #2: The authors have successfully addressed all my concerns in the revised manuscript. Hence I recommend the acceptance of this paper.

7. PLOS authors have the option to publish the peer review history of their article (what does this mean?). If published, this will include your full peer review and any attached files.

Reviewer #1: **Yes: **J. Abbas

Reviewer #2: No

---

## [Editor Report · Acceptance letter]

14 Jun 2021

PONE-D-21-07803R1 

Modelling smallholder farmers' preferences for soil fertility management technologies in Benin: A stated preference approach 

Dear Dr. Zossou:

I'm pleased to inform you that your manuscript has been deemed suitable for publication in PLOS ONE. Congratulations! Your manuscript is now with our production department. 

Kind regards, 

on behalf of

Dr. Prof. Carlos Alberto Zúniga-González 

Academic Editor

PLOS ONE